# Epigenetic Silencing of P-Element Reporter Genes Induced by Transcriptionally Active Domains of Constitutive Heterochromatin in *Drosophila melanogaster*

**DOI:** 10.3390/genes14010012

**Published:** 2022-12-21

**Authors:** Giovanni Messina, Emanuele Celauro, Renè Massimiliano Marsano, Yuri Prozzillo, Patrizio Dimitri

**Affiliations:** 1Dipartimento di Biologia e Biotecnologie “Charles Darwin”, Sapienza Università di Roma, 00185 Roma, Italy; 2Dipartimento di Bioscienze, Biotecnologie e Ambiente, Università di Bari, 70125 Bari, Italy

**Keywords:** transposable elements, P-elements, constitutive heterochromatin, position effect variegation (PEV), *Drosophila melanogaster*

## Abstract

Reporter genes inserted via P-element integration into different locations of the *Drosophila melanogaster* genome have been routinely used to monitor the functional state of chromatin domains. It is commonly thought that P-element-derived reporter genes are subjected to position effect variegation (PEV) when transposed into constitutive heterochromatin because they acquire heterochromatin-like epigenetic modifications that promote silencing. However, sequencing and annotation of the *D. melanogaster* genome have shown that constitutive heterochromatin is a genetically and molecularly heterogeneous compartment. In fact, in addition to repetitive DNAs, it harbors hundreds of functional genes, together accounting for a significant fraction of its entire genomic territory. Notably, most of these genes are actively transcribed in different developmental stages and tissues, irrespective of their location in heterochromatin. An open question in the genetic and molecular studies on PEV in *D. melanogaster* is whether functional heterochromatin domains, i.e., heterochromatin harboring active genes, are able to silence reporter genes therein transposed or, on the contrary, can drive their expression. In this work, we provide experimental evidence showing that strong silencing of the *Pw^+^* reporters is induced even when they are integrated within or near actively transcribed loci in the pericentric regions of chromosome 2. Interestingly, some *Pw^+^* reporters were found insensitive to the action of a known PEV suppressor. Two of them are inserted within *Yeti*, a gene expressed in the deep heterochromatin of chromosome 2 which carries active chromatin marks. The difference sensitivity to suppressors-exhibited *Pw^+^* reporters supports the view that different epigenetic regulators or mechanisms control different regions of heterochromatin. Together, our results suggest that there may be more complexity regarding the molecular mechanisms underlying PEV.

## 1. Introduction

Transposable elements (TEs) represent a conspicuous fraction of eukaryotic genomes, varying from 3% in yeast, 15% in *D. melanogaster*, 45% in humans, over 50% in maize and up to 90% in other plants [1,2,3,4,5].

Based on their structure and transposition mechanism, eukaryotic TEs are classified into two main classes. Class I contains retrotransposons that are mobilized via RNA transposition intermediates, whereas class II elements transpose either using a “cut and paste” mechanism (Subclass I), rolling-circle replication [6], or single strand excision followed by extrachromosomal replication [7].

TEs have shaped eukaryotic genomes [8] including regions of constitutive heterochromatin where a build-up of TEs has been documented in many evolutionarily distant organisms [9,10].

Constitutive heterochromatin has been originally defined based on cytological features, i.e., the compact state during all phases of the cell cycle and C-banding staining. In addition, low gene density, low rate of meiotic recombination, high content of repetitive DNAs and specific epigenetic signatures are hallmarks of the constitutive heterochromatin [11]. However, the lack of functional genes is no more considered a general feature of constitutive heterochromatin of *D. melanogaster* and other species [11]. Indeed, genome sequencing and annotation have shown that this peculiar genomic compartment contains hundreds of transcriptionally active genes, both unique and repetitive [12,13,14], which together account for a large fraction of the genomic territory of *D. melanogaster* constitutive heterochromatin [11]. 

Among the *Drosophila* TEs that have been extensively characterized both at the genetic and molecular levels, P and I elements are naturally occurring transposons which have been found to insert themselves into the constitutive heterochromatin of *D. melanogaster* and eventually target the genes therein located [15,16,17,18,19,20,21,22]. 

Engineered P-elements have been routinely used as DNA integration vectors or to generate collections of single-element insertion lines useful for both forward and reverse genetic studies of gene disruption, enhancer trapping and protein trapping [23,24,25,26,27,28,29,30,31,32].

Beside the above applications, engineered P-elements represent useful tools to monitor the functional state of diverse chromatin domains of *D. melanogaster* genome. When inserted into regions of constitutive heterochromatin, P-associated reporters undergo strong silencing or expressed following a mosaic pattern, in which some cells fully or partially display the reporter phenotype (known as variegated phenotype), while other cells do not [33,34,35]. The variegated phenotypes produced by transgene insertions into constitutive heterochromatin are very similar to those usually ascribed to position effect variegation (PEV) caused by chromosome rearrangements that relocate a euchromatic gene nearby to constitutive heterochromatin [36]. It is generally believed that the PEV phenotypes displayed by a variety of P-element-reporter genes reflect the silenced state of a given heterochromatin region where the elements are inserted. The variegated phenotype of P-reporter genes inserted in constitutive heterochromatin can be suppressed by known genetic modifiers of PEV [21] and by the Y chromosome, a known suppressor of PEV [37].

P-element-based vectors containing the *mini-white* gene driven by an hsp70 promoter were also used to characterize the structure of chromatin domains of the *D. melanogaster* fourth chromosome [38,39]. The results of these works have shown that this chromosome contains heterochromatic domains showing a variegating phenotype alternating with euchromatic domains showing a fully pigmented eye phenotype.

The resistance to exogenous nuclease digestion [40] and to methylation [41] exhibited by P-element reporters inserted within constitutive heterochromatin and their relatively ordered nucleosome arrays [42] together suggested that heterochromatic silencing is due to the loss of sites for binding of transcription factors and/or RNA polymerase. 

In conclusion, based on the aforementioned results, P-element reporter genes have been considered a useful tool to investigate the genetic and molecular bases underlying the PEV phenomenon.

Histone modifications have been shown to play a role in the occurrence of epigenetic silencing and thus contribute to PEV [36]. Notably, actively transcribed genes in constitutive heterochromatin must be accessible to RNA pol II and cis-acting transcriptional regulators, although they have regulatory requirements different from those of euchromatic genes [43]. In particular, combinations of negative and active histone modification marks, together with the contribution of key epigenetic regulators, such as the HP1 and SU(VAR)3-9 proteins, are likely to contribute to establishing gene expression in constitutive heterochromatin [11,44,45,46,47,48,49,50,51,52].

Here, we asked whether a given P-element reporter gene inserted into constitutive heterochromatin, within or nearby a transcribed gene, would “sense” the active chromatin state of the surrounding domain, with its expression being driven by the regulatory elements of that region or, on the contrary, would it undergo PEV. To answer this question, we examined 12 lines each containing a *mini-white* reporter gene carried by a P-element (from here indicated as *Pw^+^* reporters) cytologically and molecularly mapped to the pericentric heterochromatin of chromosome 2 (Table 1 [21,35,53]).

Our results provide new hints on the mechanisms of epigenetic silencing induced by constitutive heterochromatic in *D. melanogaster*.

## 2. Materials and Methods

### 2.1. Drosophila Strains

Fly cultures and crosses were carried out at 25 °C in standard cornmeal yeast medium. The KV lines, each carrying a single *SUPor-P* element [35], were gifts of Robert Levis. The *Su(var)205^2^* and *aubergne^QC42^* mutants were gifts of Sarah Elgin, while the *Su(var)30^71^* and *Su(var)3-9^1^* were gifts of Gunter Reuter. The *LP1* mutant was generated by Messina et al. [53].

### 2.2. Mapping of the P White+ (Pw^+^) Reporters 

The flanking sequences of the KV lines were obtained by inverse PCR (done by Robert Levis) and their locations are reported in FlyBase. The mapping of *19.74.3* and *LP1* lines was reported by Messina et al. [53].

### 2.3. Complementation Test with Df(2Rh)MS41-10 

*Pw^+^/SM1, Cy* females were crossed *to Df(2Rh)MS41-10/SM1, Cy* males, and the progeny was scored for the presence of *Cy^+^* and *Cy* flies. The expected ratio between *Cy* and *Cy^+^* among the offspring of these crosses was 2:1 because *Cy* is homozygous lethal. The absence of the *Cy^+^* progeny indicated that a given *Pw^+^* insert was lethal over the deficiency. *KV00363* and *LP1* were found to be lethal over *Df(2Rh)MS41-10* (all progeny showing the *Cy* phenotype), while for the other inserts the ratio between *Cy* and *Cy^+^* phenotypes was close to 2:1. For each complementation test, over 100 adults were counted.

### 2.4. Genetic Crosses with PEV Suppressors

We crossed w/w; *Su(var)205^2^/SM6, CyS*; +/+ or *w*/*w*; +/+; *Su(var)30^71^/TM3, Sb* females to males *w/Y; Pw^+^/SM1,Cy* from 10 selected lines carrying a *Pw^+^* reporter mapped to 2R heterochromatin balanced over the SM1,Cy chromosome (Figure 1). For the crosses in Figure 1A, we isolated the F1 male progeny carrying the suppressor (*w/Y; Pw^+^/Suvar*) and the control progeny carrying only the variegated *Pw^+^* reporter (*w/Y; Pw^+^/SM1, Cy*). Similarly, for the crosses in Figure 1B, we isolated the F1 male progeny carrying the suppressor (*w/Y*; *Pw^+^/+*; *Suvar/+*), and the control progeny with only the variegated *Pw+* reporter (*w/Y*; *Pw^+^/+*; *TM3*, *Sb/+*). All the aforementioned male progeny were isolated for eye pigment quantification.

### 2.5. Eye Pigment Assays

The extraction of red eye pigment was performed according to Ephrussi and Herold [55], as described in Prozzillo et al. [56]. For each genotype, three replicate samples of 10 heads were performed. Absorbance at 480 nm was then measured using a 96-well plate in a VICTOR Multilabel Plate Reader spectrophotometer (PerkinElmer, Waltham, MA, USA). Photographs of representative adult fly eyes were taken using a Nikon SMZ745T stereoscopic microscope (Minato, Tokyo, Japan) equipped with a digital C-mount camera.

### 2.6. Distribution of Epigenetic Marks

The distribution of the epigenetic marks in the regions of constitutive heterochromatin is available in Genome Browser of FlyBase and was also retrieved from the ModEncode data.

## 3. Results

### 3.1. Analysis of P-Element Insertions

First, we asked whether *Pw^+^* reporters inserted into actively transcribed domains of constitutive heterochromatin are subjected to PEV. To answer this question, among a collection of KV lines [21] showing strong *mini-white* silencing, we selected 10 lines with elements located within the pericentric portions of chromosome 2 (Figure 2). These regions have been extensively characterized at both genetic and molecular levels [11,14,20,43,53,57,58,59,60,61]. Each KV line carries a single *SUPor-P* element which has two reporter genes, the *mini-white* gene and the intron-less *yellow* gene, as well as two “suppressor of hairy wing” [*Su(Hw)*] binding sites flanking the *mini-white* gene [62]. This design allows a robust expression of the *yellow* reporter and, together with the presence of the Y chromosome, facilitates the recovery of *SUPor-P* pericentric insertions that would be otherwise missed due to the strong silencing of the *mini-white* reporter [21,63].

The analysis of the insertion sites showed that the *SUPor-P* element are located within or close to genes that are expressed (Table 1). In particular, as shown in Figure 3, according to FlyBase and FlyAtlas, the genes linked to the inserted elements are expressed during different developmental stages and in the eyes of both male and female. We also examined the *LP1* line carrying a *Pw^+^* reporter into the *Yeti* gene and its progenitor insert of line *19.74.3*, which maps about 2 kb downstream *Yeti* ([53], Table 1). In addition to the *mini-white* reporter, these elements contain Fab-7 insulator sequences characteristic of the *bithorax* complex [64].

All the P-insertions mapping to the right arm of chromosome 2 heterochromatin (2Rh) were tested in complementation analyses with *Df(2Rh)MS41-10*. This deletion lacks almost the entire mitotic heterochromatin of 2Rh, spanning from the deep region h40 to the very distal 46 [58,59,65], together with all the known genes, including *rolled, CG40191, CG41265, Yeti, dpr21, Haspin* and *p120 ctn* (See Figure 1, from Marsano et al. [11]). These results showed that, with the exception of *LP1* and *KV00363* (lethal alleles of *Yeti*), the analyzed P-inserts in *2Rh* were fully viable over *Df(2Rh)MS41-10*. The insert in line *KV00462* which maps to *2Lh* was homozygous viable. Thus, it is conceivable that, while the expression of *Yeti* was clearly affected, the expression of the genes linked to the other insertions was not significantly perturbed. Indeed, in the *LP1* homozygotes no *Yeti* transcripts were detected [53].

Together, from these analyses it emerged that, despite their location in expressed domains of constitutive heterochromatin, all the examined lines exhibited a strong silencing of the *Pw^+^* reporters. The males of some lines (*KV00524, KV00158, KV00384, KV00376* and *19.74.3*) exhibited a slightly stronger red-eye pigmentation compared to that shown by the females of the same line (Figure 2), an effect that could be due to the PEV suppression exerted by the Y chromosome [36,37].

### 3.2. Experiments with Genetic Modifiers of PEV

Loss-of-function mutations resulting in dominant suppression of PEV were shown to identify crucial epigenetic regulators, such as the HP1 protein [36]. Thus, first we investigated whether the variegation of *Pw^+^* reporters showed by the lines under investigation can be suppressed by *Su(var)205* and *Su(var)307*, two dominant PEV suppressors [36]. To do this, we crossed *w/w; Su(var)205^2^/SM6, Cy,S; +/+* or *w/w; +/+; Su(var)3-7^1^/TM3, Sb* females to males of 10 selected *Pw^+^* reporters (Figure 2) balanced over the *SM1, Cy* chromosome (Figure 1; Section 2). 

The results of this analysis are summarized in Figure 4. *Su(var)205^2^* clearly suppressed the variegation of *Pw+* reporters in lines *KV00158, KV00376, KV00369* and *KV00171*. However, this was not the case for the remaining five *Pw^+^* reporters in lines *KV00249, KV00299, KV00363, KV00524* and *LP1*, which were quite insensitive to the suppression. *Su(var)307^1^* suppressed most *Pw^+^* reporters, although with a variable efficiency. However, the reporters of *LP1* and *KV00299* lines showed only a slight suppression effect, while the one of *KV00363* was clearly insensitive.

Moreover, mutations in *piwi, aubergine* or *homeless* genes, encoding components of the piRNA pathway [66,67], also suppress the silencing of *Pw^+^* inserted in heterochromatin as a result of reduction of H3 Lys9 methylation and delocalization of HP1. Thus, we also tested the effect of the *aubergine^QC42^* mutant allele on the variegation of the *Pw^+^* reporters under investigation (see Section 2 for the scheme of crosses). As shown in Figure 4, the trend of the effect was very similar to that found with *Su(var)205^2^* in that the *Pw^+^* reporters of *KV00249, KV00299, KV00363* and *LP1* lines were insensitive to suppression, with the exception of *KV00524*.

The product of the wild-type *Su(var)205* gene, the heterochromatin-associated HP1 protein, interacts with the histone methyltransferase (HMTase) encoded by the wild-type *Su(var)3-9* gene. The SU(VAR)3-9 methyltransferase selectively methylates histone H3 at lysine 9 (H3-K9), and it is generally accepted that H3K9me stabilizes the binding of HP1a to chromatin [68,69,70]. The *Su(var)3-9* mutant alleles are known to be dominant modifiers of PEV [36]. We then tested the effect of the *Su(var)3-9^1^* mutation on the variegation of *Pw^+^* reporters from *KV00249*, *KV00299*, *KV00363*, *LP1* and *19.74.3* lines. As shown in Figure 5, the variegation of *Pw^+^* reporters in *KV00249*, *KV00299*, *KV00363* and *LP1* lines was insensitive to suppression by *Su(var)3-9^1^*, whereas the *Pw^+^* variegation in the *19.74.3* was efficiently suppressed. In conclusion, using different PEV suppressors (*Su(var)205^2^*, *Su(var)3-9^1^* and *aubergine^QC4^*), we obtained comparable results with the same *Pw^+^* reporters (Figure 4 and Figure 5). 

## 4. Discussion

It is generally believed that PEV is induced by transcriptionally inactive constitutive heterochromatin. The aim of this work was to further investigate this aspect by studying the heterochromatin-induced silencing on a number of *Pw^+^* reporters inserted into different positions of the pericentromeric regions of chromosome 2, within or nearby a transcribed gene (Table 1; Figure 3 and Figure 6). We found that all the examined reporters subjected to strong PEV (Figure 2) map within or close to genes/domains that support an appreciable level of transcription during development and in the eye (Figure 3). Moreover, some reporters were suppressed by classical dominant PEV suppressors encoding epigenetic factors/regulators required for heterochromatin establishment, while other reporters in lines *KV00249*, *KV00299*, *KV00363* and *LP1* were less sensitive or even insensitive to the action of the suppressors (Figure 4, Figure 5 and Figure 6). In general, there appears to be no significant correlation between the *mini-white* expression levels for each *Pw^+^* reporter line and the transcription levels of the heterochromatic gene(s) linked to it. For example, CG40191 is on average more expressed than *dpr21*, and yet the respective inserts have the same behavior towards suppression.

Although the heterogeneity of the genetic background might partially contribute to explaining the different sensitivity to epigenetic regulators tested, these results suggest that the chromatin state characteristic of transcriptionally active heterochromatin may not be sufficient to guarantee the proper expression of a given reporter gene inserted within or nearby. In other words, the regulatory requirements for the expression of the autochthonous genes in heterochromatin seem to be different, or even antipodal, from those needed for adventitious DNA sequences, such as the *Pw^+^* reporters. However, the observation that the silencing of some reporters is abolished by the tested suppressors also suggests that the product of wild-type alleles of some suppressors, on one hand, can induce the silencing of the reporters, but, on the other hand, may be required for, or is compatible with, the expression of the genes located in pericentric heterochromatin. The effect of the product of wild-type alleles of suppressors could be performed directly or indirectly. 

Indeed, the expression of *light* and *rolled* heterochromatic genes of chromosome 2 was found to be affected when moved away from their native location to euchromatin by chromosome rearrangements [70,71]. Moreover, genetic and molecular studies have shown that the proper expression of *light, rolled,* and other heterochromatic genes such as *Rpl15* and *Dbp80* depends on the HP1a protein [72,73,74,75]. In accordance, large-scale mapping experiments carried out in *Drosophila* Kc cells have shown that HP1a and SU(VAR)3-9 are associated with pericentric genes, such as *light* and *rolled* genes [44,46,47]. 

Together, the available findings support the idea that histone marks, together with HP1a and other epigenetic regulators, are involved to ensure the expression of genes located in constitutive heterochromatin. However, some apparently contradictory results [44,51] indicated that differences in the epigenetic regulation of heterochromatic genes may exist between in vivo and in vitro systems.

The insensitivity to suppressors showed by *Pw^+^* reporters in *KV363* and *LP1* lines is intriguing for several reasons. Firstly, *Yeti* is an efficiently expressed gene in the heterochromatin of chromosome 2 (Figure 2 [11,53,76,77,78,79]) and carries active chromatin marks (Table 1). Secondly, the P-insertion of the *19.74.3* line, the progenitor of *LP1*, maps roughly 2 kb downstream the 5′ of *Yeti* (Figure 6), but its silencing is efficiently suppressed by *Su(var)205^2^*, *Su(var)307^1^*, *Su(var)309^2^* and *aubergne^QC42^* (Figure 4 and Figure 5). Thirdly, *Su(var)205*, *Su(var)309* and *Su(var)307* are among the most general and effective PEV suppressors and were already found to efficiently erase the silencing of many pericentric reporters [36,80]. Finally, the distribution of HP1 and SU(VAR)309 proteins along the pericentric regions was found to be enriched in the heterochromatin of 2R where *Yeti* is located [44,46,47].

The molecular bases underpinning insensitivity to suppressors exhibited by *Pw^+^* reporters in *KV363* and *LP1* lines are currently elusive. A trivial explanation for the apparently strong silencing could be that the *Pw^+^* element in these lines is structurally defective so that the expression of the *mini-white* is compromised. However, we found that in both lines the element can still transpose in the presence of the *Delta-2-3* element, a known source of the P-transposase, giving rise to de novo fully pigmented red-eye insertions (not show). This result excludes the occurrence of significant DNA sequence alterations in the *Pw+* element of both *KV363* and *LP1* lines. 

It is possible that the chromatin state of the *Yeti* gene is regulated by epigenetic regulators/mechanisms different from those controlled by the wild-type alleles of the tested PEV suppressors.

We did not find recurrent repeated sequences flanking the reporters, suggesting the absence of specific relationships between location and sensitivity to suppression. However, two non-coding RNAs, *CR44042* and *CR44043*, were associated with *Yeti* and *l(2)41Ab* genes (Figure 6), respectively, with the *Pw^+^* reporter of *KV00299* being inserted within *CR44043*. It could be possible that the transcription of *CR44042* and *CR44043* may interfere somehow with that of the reporters.

A consistent body of experimental evidence suggested that nascent RNA may play a direct role in transcription and chromatin regulation, in that it may function as regulator of its own expression, giving rise to positive-feedback loops in which active gene expression states can be maintained [81]. This model could explain the resistance to all PEV suppressor exhibited by the *Pw^+^* reporters in *LP1* and *KV00363* lines. We hypothesize that in wild-type conditions the *Yeti*-encoded RNA interacts with its own gene at the level of DNA/chromatin, acting as an epigenetic insulator or as a positive effector, giving rise to an active chromatin state which allows a stable expression of *Yeti*. However, following a P-insertional mutation, a consequent lack of functional *Yeti* transcripts (or a low level of transcription) would shut-down the expression of the gene disrupting its active chromatin state with the acquisition of a closed chromatin structure. As a consequence, the *Pw^+^* reporters inserted within *Yeti* would become silenced (Figure 7). Such positive RNA feedback mechanism, if existing, should be insensitive to the action of classical PEV suppressors.

## 5. Conclusions

In conclusion, we have shown that transcriptionally active domains of constitutive heterochromatin can induce epigenetic silencing of *Pw^+^* reporters. Whatever the genetic bases of this phenomenon are, our results represent a remarkable paradox suggesting that there may be more complexity regarding the molecular mechanisms underlying PEV.

## Figures and Tables

**Figure 1 genes-14-00012-f001:**
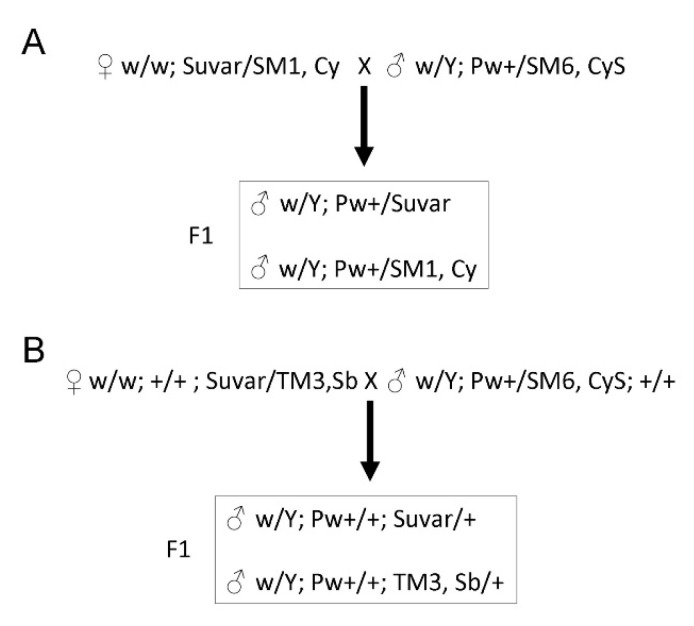
Scheme of genetic crosses. (**A**) Crosses with PEV suppressor mapping to chromosome 2 (*Su(var)205^2^* and *aubergne^QC4^*); (**B**) Crosses with PEV suppressor mapping to chromosome 3 (*Su(var)30^71^* and *Su(var)3-9^1^*). For a detailed explanation of the crosses see Material and Methods.

**Figure 2 genes-14-00012-f002:**
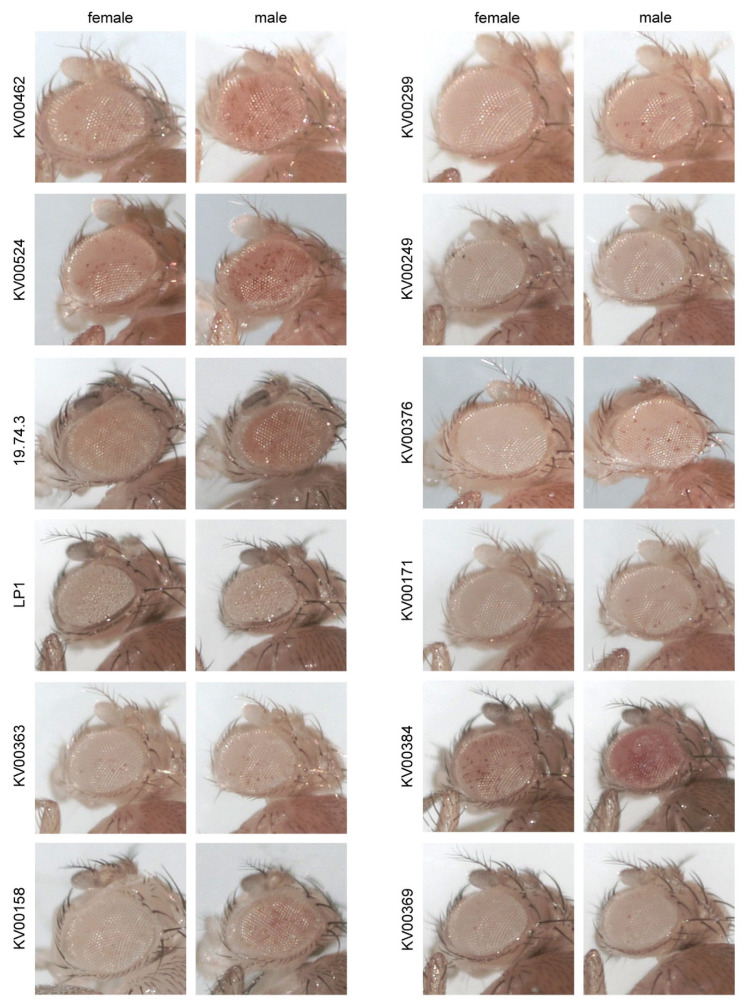
Strong silencing of *Pw^+^* reporters. Phenotypic analysis of the 12 lines carrying a single *Pw^+^* reporter inserted in constitutive heterochromatin of chromosome 2. All the *Pw^+^* reporters are balanced over the *SM1, Cy* chromosome and exhibited a strong silencing giving rises to extremely variegated eyes which in some cases are almost white. The males of lines *KV00462, KV00524, 19.74.3 KV00158* and *KV00384* exhibited a stronger red-eye pigmentation compared to that of females, a difference that could be due to the presence of Y chromosome.

**Figure 3 genes-14-00012-f003:**
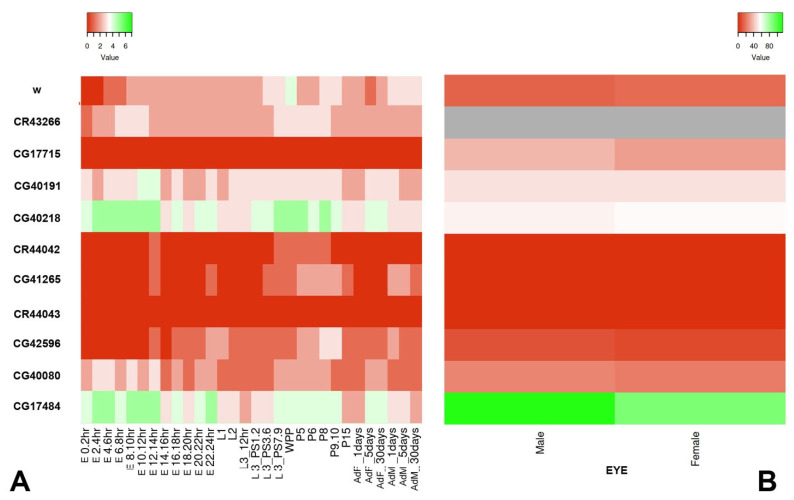
Heatmaps showing expression profile of the heterochromatic genes linked to the examined *Pw^+^* reporters compared to that of *white* gene. Expression across developmental stages (**A**) and in the eyes of males and females (**B**). Shades of color from red to green indicate the expression bin classification from 1 (no/extremely low expression) to 7 (very high expression). The gray color means that no data about *CR43266* expression are available. Developmental stage and eye expression data were obtained from FlyBase and FlyAtlas, respectively.

**Figure 4 genes-14-00012-f004:**
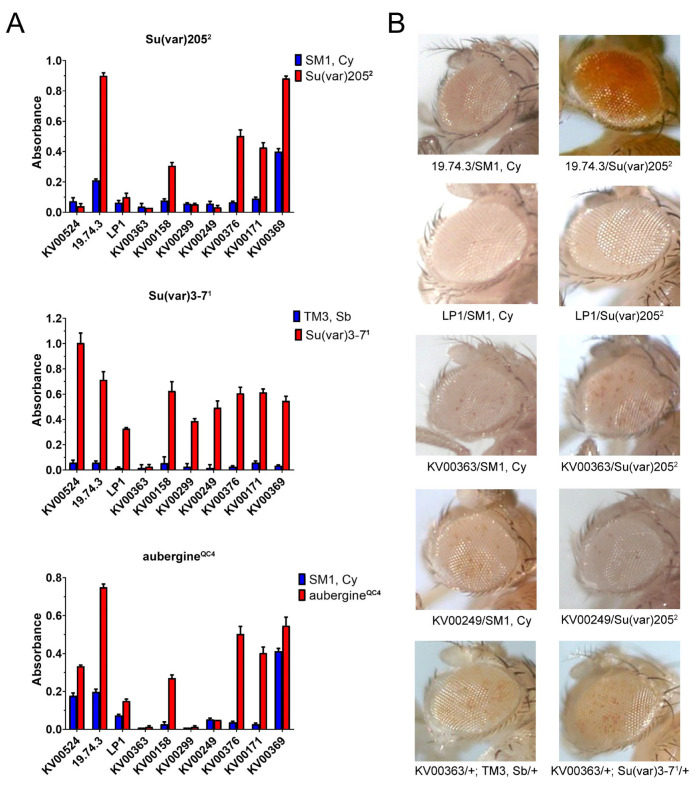
Effects of dominance of *Su(var)205^2^*, *Su(var)3-7^1^* and *aubergine^QC4^* on the silencing of *Pw^+^* reporters. (**A**) Histograms summarizing the quantification of eye pigment in presence and absence of PEV suppressors; (**B**) Examples of eye phenotypes of *Pw^+^* reporters in flies with or without suppressors. The silencing of *Pw^+^* reporter in *19.74.3* line was efficiently suppressed by *Su(var)205^2^*, *Su(var)3-7^1^* and *aubergine^QC4^*; silencing in *KV00363* and *LP1* lines was unaffected by *Su(var)205^2^* and *aubergine^QC4^*, while silencing in *LP1* was only partially suppressed by *Su(var)3-7^1^*.

**Figure 5 genes-14-00012-f005:**
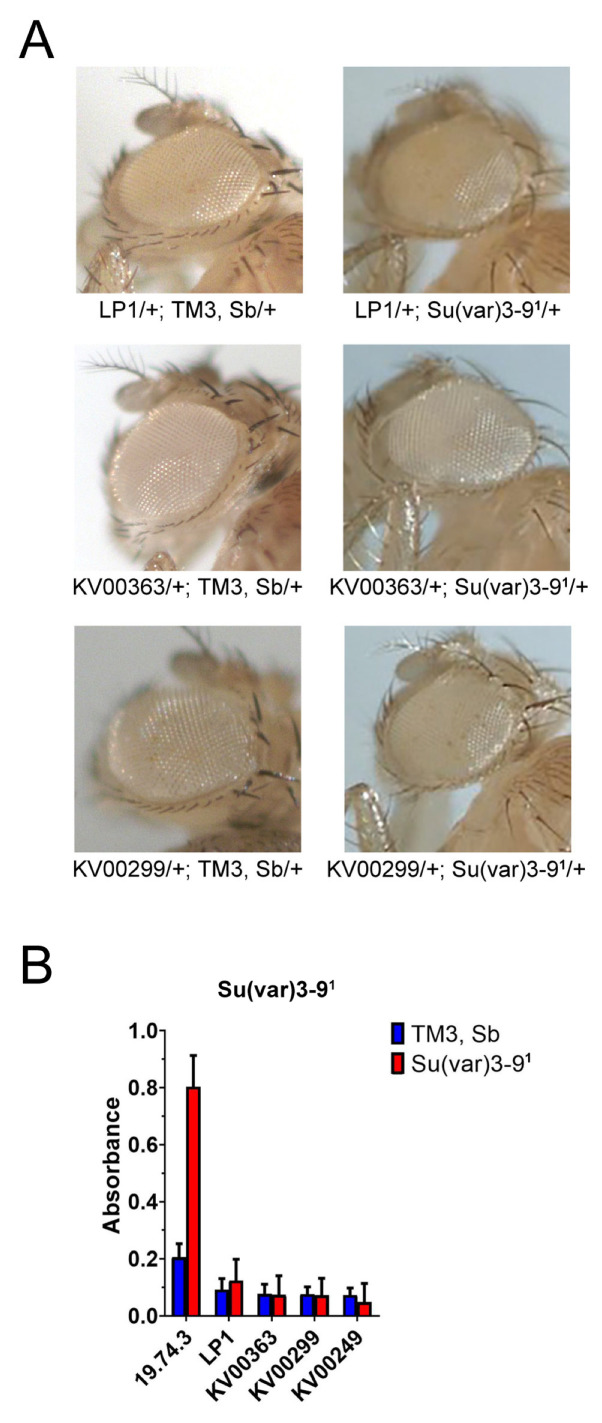
Effects of *Su(var)3-9^1^* on silencing of *Pw^+^* reporters. (**A**) Examples of eye phenotypes of *Pw^+^* reporters in flies with or without *Su(var)3-9^1^*. (**B**) Histograms summarizing the quantification of eye pigment in presence and absence of *Su(var)3-9^1^.* The silencing of the *Pw^+^* reporter in *19.74.3* was efficiently suppressed, while that of *KV00363, LP1, KV00299* and *KV00249* was unaffected. For the scheme of crosses see Figure 1 and Materials and Methods.

**Figure 6 genes-14-00012-f006:**
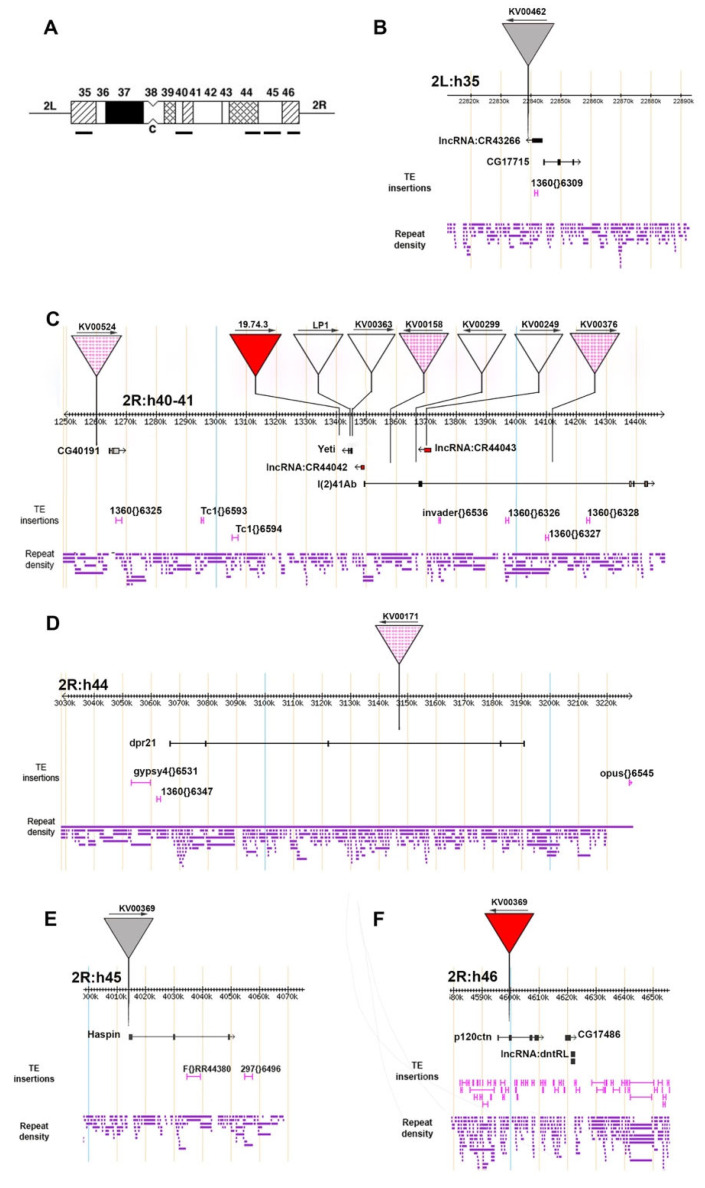
Genomic localization of the *Pw^+^* reporters under investigation. (**A**) Cytogenetic map of the heterochromatin of chromosome 2 showing the location of the genomic regions analyzed in this work. From left to right: 2L:h35, 2R:h40-41, 2R: h44, 2R:h45-46, 2R:h46. The genomic regions in (**B**–**F**) show the localization of the *Pw^+^* reporter insertions relative to genes and the distribution of TEs and repetitive sequences. Red triangle = fully suppressed *Pw^+^* reporter; dashed triangle = *Pw^+^* reporter suppressed only in some cases; white triangle = *Pw^+^* reporter insensitive to suppression; grey triangle = *Pw^+^* reporter not tested for suppression.

**Figure 7 genes-14-00012-f007:**
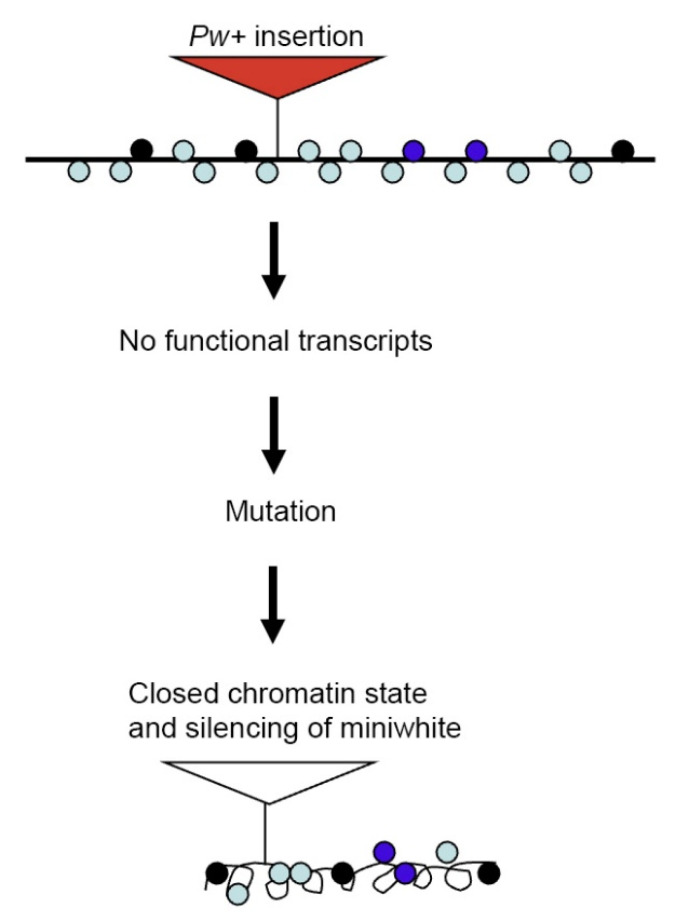
*Pw^+^* silencing by perturbation of a positive RNA feedback mechanism. The insertion of a *Pw+* reporter within the *Yeti* gene affects its expression. According to a positive RNA feedback model, the consequent lack of functional *Yeti* transcripts would induce the formation of a closed chromatin structure of the gene itself. As a result, the *mini-white* gene is silenced. Colored circles represent different chromatin marks.

**Table 1 genes-14-00012-t001:** Features of the insertion lines described in this work.

Line	P-Element	P-Element Position	Affected Gene(s)	Gene Position	Insert Position Relative to Nearby Genes	Sensitivity to Suppressors	Chromatin State (Gene)	Chromatin State (Insertion Site)	Natural TE Insertions (+/− 5 kb)
*KV00462*	*P{SUPor-P}*	2L:22839699 [−]	*lnRNA:CR43266* *CG17715*	2L:22840517..22843668 [−] 2L:22844220..22854541 [+]	818 bp downstream 4521 bp upstream	Y	1 1,2,7	7	1360
*KV00524*	*P{SUPor-P}*	2R:1260501 [+]	*CG40191*	2R:1264140..1267747 [+]	3639 bp upstream	Y/N	1,7	7	1360
*19.74.3*	*P{Fab7-w#un1}*	2R:1341511 [+]	*Yeti (CG40218)*	2R:1343403..1345119 [−]	1892 bp upstream	Y	1	NR	ND
*LP1*	*P{Fab7-w#un1}*	2R:1345084 [+]	*Yeti (CG40218)*	2R:1343403..1345119 [−]	Start codon	N	1	1	
*KV00363*	*P{SUPor-P}*	2R:1345097 [+]	*Yeti (CG40218)*	2R:1343403..1345119 [−]	5′UTR	N	1	1	
*KV00158*	*P{SUPor-P}*	2R:1358001 [−]	*l(2)41Ab (CG41265)*	2R:1349392..1443935 [+]	1st intron	Y/N	NR	NR	invader1 1360 (3 copies)
*KV00299*	*P{SUPor-P}*	2R:1366708 [+]	*l(2)41Ab* *(CG41265)*	2R:1349392..1443935 [+]	1st intron	N	NR	NR	
*KV00249*	*P{SUPor-P}*	2R:1370175 [+]	*l(2)41Ab* *(CG41265)* *lnRNA:CR44043*	2R:1349392..1443935 [+] 2R:1368921..1371468 [−]	2nd intron Gene body	N	NR ND	NR	
*KV00376*	*P{SUPor-P}*	2R:1412310 [+]	*l(2)41Ab* *(CG41265)*	2R:1349392..1443935 [+]	2nd intron	Y/N	NR	NR	
*KV00171*	*P{SUPor-P}*	2R:3147095 [−]	*dpr21* *(CG42596)*	2R:3066499..3191011 [+]	Intron 5	Y/N	ND	ND	Gypsy4 1360
*KV00384*	*P{SUPor-P}*	2R: 4014047 [+]	*Haspin* *(CG40080)*	2R:4014111..4049342 [+]	64 bp upstream	Y	1	NR	F 297
*KV00369*	*P{SUPor-P}*	2R:4593333 [−]	*p120ctn* *(CG17484)*	2R:4595288..4609492 [+]	1955 bp upstream	Y	1,2,7	7	Doc3 (2 copies) Doc 1360 (5 copies) INE-1 (5 copies) Diver (2 copies) Cr1a Burdock Gypsy12

Genomic positions are relative to the dm6 genome assembly. [+] and [−] indicate the orientation of genes and inserts relative to the genome assembly. The chromatin state detected in the gene body and at the P-element insertion site are described in Kharchenko, et al. [54] (chromatin state 1: Active promoter/transcription start site region; chromatin state 2: Actively transcribed exon; chromatin state 7: Heterochromatin). 2L: Heterochromatin of the left arm of chromosome 2; 2R: Heterochromatin of the right arm of chromosome 2. NR: Not Reported. ND: None Detected. Y: Suppressed; N: Not Suppressed.

## Data Availability

Data are contained within the article.

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
