# Peer review of "Epigenetic Silencing of P-Element Reporter Genes Induced by Transcriptionally Active Domains of Constitutive Heterochromatin in Drosophila melanogaster"

_genes, 2022, doi:10.3390/genes14010012_

Round 1

Reviewer 1 Report

This interesting paper shows that the integration of Pw+ reporters within or near actively transcribed loci in the pericentric regions is insufficient to actively transcribed them. Indeed, they are strongly silenced. The message is important and supported by the experiments. However, I will comment on specific weak points that need to be changed for the correct data interpretation.

1-         The interpretation regarding the expression of the Pw+ reporters is valid only if reporter P elements integration does not modify the chromatin organization/transcription of the "actively transcribed loci." This control is essential when comparing the three different insertions in the Yeti loci giving rise to different results. Therefore, quantitative transcriptional analyses of the heterochromatic-gene(s) linked to the examined Pw+ reporters (described in table 1) need to be determined (for instance, by qPCR in the total head of control flies vs. Pw+ line flies). 

2-         It has been previously shown that the variegation effect is highly susceptible to the background (doi: 10.1186/s13072-019-0314-5). However, in this study, this is not considered for interpretation. This effect is observed, for instance, when comparing the absorbance of control lines between figure 4 A (blue bars - balancers SM1 vs. TM3). Unless multiple crosses against a control line have unified the background of used Pw+ lines before their white expression analyses, this point must be discussed in the "Discussion" section. Could the authors clarify whether the background might interact with the dominant mutants (su(var)2052, aubergneQC42, su(var)3-71, su(var)3-91), partially explaining the different sensitivity to epigenetics regulators?

3-         Could the authors discuss (in the discussion section) the correlation or no correlation between the white expression levels for each Pw+ reporter line (shown in figure 1) and the transcription levels of the heterochromatic-gene(s) linked to it? Thus, this is particularly informative since the reporters show different silencing effects and the associated genes' different transcriptional expression levels in the eyes.

Author Response

ANSWERS TO REVIEWER 1

1-         The interpretation regarding the expression of the Pw+ reporters is valid only if reporter P elements integration does not modify the chromatin organization/transcription of the "actively transcribed loci." This control is essential when comparing the three different insertions in the Yeti loci giving rise to different results. Therefore, quantitative transcriptional analyses of the heterochromatic-gene(s) linked to the examined Pw+ reporters (described in table 1) need to be determined (for instance, by qPCR in the total head of control flies vs. Pw+ line flies). 

Sure, this point is important. For this reason, we added new data on the viability of the Pw+ insertions  over a large deletion of 2R heterochromatin. As shown in table… all the P-insertions were used in complementation test with Df(2Rh)MS41-10 (also named MS2-10) a deletion of almost the entire mitotic heterochromatin of 2Rh, spanning from the deep region h40 to the very distal 46 (Dimitri, 1991; Corradini et al., 2003; Rossi et al., 2007). This deletion lacks almost all the known 2Rh genes, including rolled, CG40191, CG41265, Yeti, dpr21, Haspin and p120ctn (See Figure 1, from Marsano  et al., 2019). These results of these tests showed that, with the exception of KV00363 and LP1 (lethal alleles of Yeti), all the insertion are fully viable over Df(2Rh) MS2-10. Thus, while the expression of Yeti is clearly affected by the insertions in lines KV00363 and LP1 (the transcripts are absent in LP1 homozygotes, Messina et al., 2014), the expression of the active genes linked to the other P-element insertions does not appear to be significantly affected.  We believe these results can answer to the point raised by the reviewer.

2-         It has been previously shown that the variegation effect is highly susceptible to the background (doi: 10.1186/s13072-019-0314-5). However, in this study, this is not considered for interpretation. This effect is observed, for instance, when comparing the absorbance of control lines between figure 4 A (blue bars - balancers SM1 vs. TM3). Unless multiple crosses against a control line have unified the background of used Pw+ lines before their white expression analyses, this point must be discussed in the "Discussion" section. Could the authors clarify whether the background might interact with the dominant mutants (su(var)2052, aubergneQC42, su(var)3-71, su(var)3-91), partially explaining the different sensitivity to epigenetics regulators?

We have clarified this aspect in the discussion.

3-         Could the authors discuss (in the discussion section) the correlation or no correlation between the white expression levels for each Pw+ reporter line (shown in figure 1) and the transcription levels of the heterochromatic-gene(s) linked to it? Thus, this is particularly informative since the reporters show different silencing effects and the associated genes' different transcriptional expression levels in the eyes.

Actually, it appears to be no correlation. For example, CG40191 is on average more expressed than dpr21 and yet the respective inserts have the same behavior towards suppression. We have now discussed this issue in the discussion.

Reviewer 2 Report

The manuscript by Messina et al. is directly related to long-standing investigations of the position effect variegation (PEV) phenomenon consisting in stochastic silencing of euchromatin genes placed within or nearby heterochromatin by chromosome rearrangements or as a part of transgenic constructs. Constitutive heterochromatin that spans a substantial proportion of the Drosophila genome appears to be not uniform. Particularly, the regulation of activity of genes embedded in it is not fully understood yet. In the present study, authors addressed the question whether the local chromatin environment at active heterochromatin genes is compatible with activity of a gene from euchromatin. They used a typical mini-white reporter gene, ensuring red pigmentation of fly eyes (on the white gene mutant background), that was delivered into the constitutive heterochromatin loci of interest by P-element transposons. The authors found that in all 12 studied cases the mini-white reporter becomes silent being inserted nearby or within active heterochromatin genes. In addition, some of these mini-white insertions cannot be activated upon mutations in the genes encoding some known heterochromatin components. The authors proposed a model explaining the latter findings.

MAJOR comments

The selection process of 12 P-element insertions used in the study is not well explained. It is not clear what was the original collection of P-element insertions and how many of them were mapped to constitutive heterochromatin. It is quite possible that the majority of P-element insertions in heterochromatin cannot be even detected after the transgenesis step due to the complete silencing of the reporter. So, was the transgenesis step performed somehow specifically to obtain the selected 12 P-element insertions, e.g. on some mutant background that is favorable for detection of the mini-white activity?

In addition, P-element in the KV00524 line seems to be located relatively far away (~36.6 kb) from the CG40191 gene. So, it is not obvious that the CG40191 gene can be affected by this P-element insertion.

It is worthwhile briefly explaining the differences between the P{Fab7-w#un1} and P{SUPor-P} P-elements used in the study, as it is not clear whether such differences might somehow affect the studied phenomenon.

The proposed model could be relatively easy tested by measuring the expression level of the affected gene (e.g. Yeti) by RT-qPCR in fly eyes/heads in animals with and without P-element insertion within the gene. Alternatively, antibodies specific to the product of the target gene could be used to evaluate the amount of the protein either in fly eyes/heads or in polytene chromosomes.

MINOR comments:

Names of all genes and mutations have to be written in italic in all Figures.

Lines 52-53: consider changing “sequencing” to “genome sequencing”

Line 106 (Title of Table 1): consider changing “insertion line” to “insertion lines”

Line 140: consider deleting an extra “using a”

Lines 145-146: consider changing “…of constitutive heterochromatin it is avaliable in…” to “…of constitutive heterochromatin is available in…”

Lines 177-178 and 269: I found quite difficult to understand the phrases “which were quite insensitive to be suppressed” and “reporter insensitive to suppression”. These P-elements are suppressed and are not activated upon mutation in the Su(var)205 gene. Consider rephrasing

Line 208: consider changing “We then ì tested…” to “We then tested…”

Lines 256 and 279: consider writing “SUVAR3-9” as “SU(VAR)3-9” for the consistency with writing of the name of the corresponding gene in the rest of the manuscript

Line 273: consider changing “active chromatin markers” to “active chromatin marks”

Line 288: consider writing “Pw+” in italic

Table 1: consider changing “2Kbp” (2 cases) to “2 kb” (as written in the rest of the manuscript)

Legends of Figures 4A and 5B: “Histograms summarizing the results” is not enough to understand what exactly is shown on the plots.

Legends of Figures 5A and 5B appear to be swapped.

Legend of Figure 6: the meaning of all colored circles on the drawing have to be explained.

Author Response

ANSWERS TO REVIEWER 2

MAJOR comments

1. The selection process of 12 P-element insertions used in the study is not well explained. It is not clear what was the original collection of P-element insertions and how many of them were mapped to constitutive heterochromatin. It is quite possible that the majority of P-element insertions in heterochromatin cannot be even detected after the transgenesis step due to the complete silencing of the reporter. So, was the transgenesis step performed somehow specifically to obtain the selected 12 P-element insertions, e.g. on some mutant background that is favorable for detection of the mini-white activity?

We clarified this point in the first part of the results.

2. In addition, P-element in the KV00524 line seems to be located relatively far away (~36.6 kb) from the CG40191 gene. So, it is not obvious that the CG40191 gene can be affected by this P-element insertion.  

There was indeed a typo (an extra “6”), and the actual distance is 3639 bp.

  1. It is worthwhile briefly explaining the differences between the P{Fab7-w#un1} and P{SUPor-P} P-elements used in the study, as it is not clear whether such differences might somehow affect the studied phenomenon.

 We have explained the difference in the first part of the results.

4. The proposed model could be relatively easy tested by measuring the expression level of the affected gene (e.g. Yeti) by RT-qPCR in fly eyes/heads in animals with and without P-element insertion within the gene. Alternatively, antibodies specific to the product of the target gene could be used to evaluate the amount of the protein either in fly eyes/heads or in polytene chromosomes.

In a previous paper (Messina et al., 2014) we already have shown that LP1, is a loss of function lethal allele of Yeti and in the LP1 homozygous larvae the Yeti-transcripts are absent. KV00363 is also a lethal allele of Yeti, it fails to complement Df(2Rh)MS41-10 and the LP1/ KV00363 trans-heterozygous are lethal.

  1. MINOR comments

We have made all the corrections suggested by the reviewer and other additional typos and slips that we have found after a careful revision of the text.

Names of all genes and mutations have to be written in italic in all Figures.

Lines 52-53: consider changing “sequencing” to “genome sequencing”

Line 106 (Title of Table 1): consider changing “insertion line” to “insertion lines”

Line 140: consider deleting an extra “using a”

Lines 145-146: consider changing “…of constitutive heterochromatin it is avaliable in…” to “…of constitutive heterochromatin is available in…”

Lines 177-178 and 269: I found quite difficult to understand the phrases “which were quite insensitive to be suppressed” and “reporter insensitive to suppression”. These P-elements are suppressed and are not activated upon mutation in the Su(var)205 gene. Consider rephrasing

Line 208: consider changing “We then ì tested…” to “We then tested…”

Lines 256 and 279: consider writing “SUVAR3-9” as “SU(VAR)3-9” for the consistency with writing of the name of the corresponding gene in the rest of the manuscript

Line 273: consider changing “active chromatin markers” to “active chromatin marks”

Line 288: consider writing “Pw+” in italic

Table 1: consider changing “2Kbp” (2 cases) to “2 kb” (as written in the rest of the manuscript)

Legends of Figures 4A and 5B: “Histograms summarizing the results” is not enough to understand what exactly is shown on the plots.

Legends of Figures 5A and 5B appear to be swapped.

Legend of Figure 6: the meaning of all colored circles on the drawing have to be explained.